# Adversarial Self-Supervised Contrastive Learning

**Minseon Kim**[1], **Jihoon Tack**[1], **Sung Ju Hwang**[1,2]
KAIST[1], AITRICS[2]
{minseonkim, jihoontack, sjhwang82}@kaist.ac.kr

## Abstract

Existing adversarial learning approaches mostly use class labels to generate adversarial samples that lead to incorrect predictions, which are then used to augment the training of the model for improved robustness. While some recent works propose semi-supervised adversarial learning methods that utilize unlabeled data, they still require class labels. However, do we really need class labels *at all*, for adversarially robust training of deep neural networks? In this paper, we propose a novel adversarial attack for unlabeled data, which makes the model confuse the instance-level identities of the perturbed data samples. Further, we present a self-supervised contrastive learning framework to adversarially train a robust neural network without labeled data, which aims to maximize the similarity between a random augmentation of a data sample and its *instance-wise* adversarial perturbation. We validate our method, *Robust Contrastive Learning (RoCL)*, on multiple benchmark datasets, on which it obtains comparable robust accuracy over state-of-the-art supervised adversarial learning methods, and significantly improved robustness against the *black box* and *unseen* types of attacks. Moreover, with further joint fine-tuning with supervised adversarial loss, RoCL obtains even higher robust accuracy over using self-supervised learning alone. Notably, RoCL also demonstrate impressive results in robust transfer learning.

## 1   Introduction

The vulnerability of neural networks to imperceptibly small perturbations [1] has been a crucial challenge in deploying them to safety-critical applications, such as autonomous driving. Various studies have been proposed to ensure the robustness of the trained networks against adversarial attacks [2–4], random noise [5], and corruptions [6, 7]. Perhaps the most popular approach to achieve adversarial robustness is adversarial learning, which trains the model with samples perturbed to maximize the loss on the target model. Starting from Fast Gradient Sign Method [8] which apply a perturbation in the gradient direction, to Projected Gradient Descent [9] that maximizes the loss over iterations, and TRADES [2] that trades-off clean accuracy and adversarial robustness, adversarial learning has evolved substantially over the past few years. However, conventional methods with adversarial learning all require *class labels* to generate adversarial attacks.

Recently, self-supervised learning [10–14], which trains the model on unlabeled data in a supervised manner by utilizing self-generated labels from the data itself, has become popular as means of learning representations for deep neural networks. For example, prediction of the rotation angles [10], and solving randomly generated Jigsaw puzzles [11] are examples of such self-supervised learning methods. Recently, instance-level identity preservation [12, 13] with contrastive learning has shown to be very effective in learning the rich representations for classification. Contrastive self-supervised learning frameworks such as [12–15] basically aim to maximize the similarity of a sample to its augmentation, while minimizing its similarity to other instances.

In this work, we propose a contrastive self-supervised learning framework to train an adversarially robust neural network *without* any class labels. Our intuition is that we can fool the model by generat-

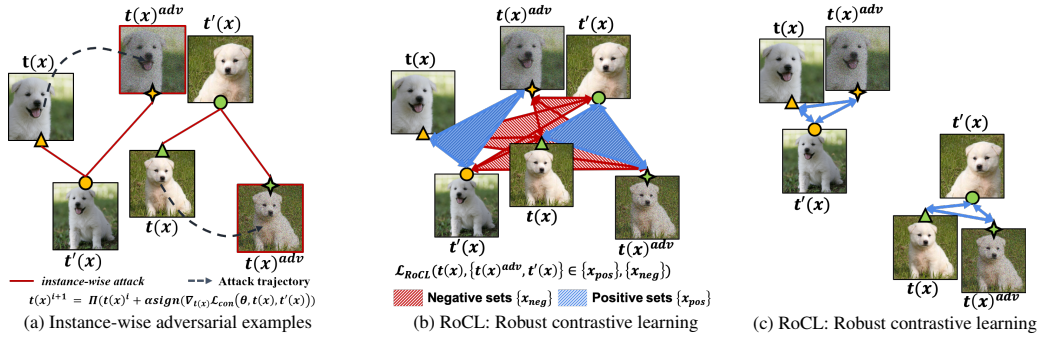

| | |
|---|---|
| (a) Instance-wise adversarial examples | (b) RoCL: Robust contrastive learning | (c) RoCL: Robust contrastive learning |

Figure 1: **Overview of our adversarial contrastive self-supervised learning.** (a) We generate instance-wise adversarial examples from an image transformed using a stochastic augmentation, which makes the model confuse the instance-level identity of the perturbed sample. (b) We then maximize the similarity between each transformed sample and their instance-wise adversaries using contrastive learning. (c) After training, each sample will have significantly reduced adversarial vulnerability in the latent representation space.

ing instance-wise adversarial examples (See Figure 1(a)). Specifically, we generate perturbations on augmentations of the samples to maximize their contrastive loss, such that the instance-level classifier becomes confused about the identities of the perturbed samples. Then, we maximize the similarity between clean samples and their adversarial counterparts using contrastive learning (Figure 1(b)), to obtain representations that suppress distortions caused by adversarial perturbations. This will result in learning representations that are robust against adversarial attacks (Figure 1(c)).

We refer to this novel adversarial self-supervised learning method as *Robust Contrastive Learning (RoCL)*. To the best of our knowledge, this is the first attempt to train robust neural networks *without any labels*, and to generate instance-wise adversarial examples. Recent works on semi-supervised adversarial learning [16, 17] or self-supervised adversarial learning [18] still require labeled instances to generate pseudo-labels on unlabeled instances or class-wise attacks for adversarial training, and thus cannot be considered as fully-unsupervised adversarial learning approaches.

To verify the efficacy of the proposed RoCL, we suggest a robust-linear evaluation for self-supervised adversarial learning and validate our method on benchmark datasets (CIFAR-10 and CIFAR-100) against supervised adversarial learning approaches. The results show that RoCL obtains comparable accuracy to strong supervised adversarial learning methods such as TRADES [2], although it does not use any labels during training. Further, when we extend the method to utilize class labels to fine-tune the network trained on RoCL with class-adversarial loss, we achieve even stronger robustness, *without* losing accuracy when clean samples. Moreover, we verify our rich robust representation with transfer learning which shows impressive performance. In sum, the contributions of this paper are as follows:

- We propose a novel **instance-wise** adversarial perturbation method which does not require any labels, by making the model confuse its instance-level identity.

- We propose a **adversarial self-supervised learning** method to explicitly suppress the vulnerability in the representation space by maximizing the similarity between clean examples and their instance-wise adversarial perturbations.

- Our method obtains **comparable** robustness to supervised adversarial learning approaches without using any class labels on the target attack type, while achieving **significantly better** clean accuracy and robustness on unseen type of attacks and transfer learning.

## 2   Related Work

**Adversarial robustness**   Obtaining deep neural networks that are robust to adversarial attacks has been an active topic of research since Szegedy et al.[1] first showed their fragility to imperceptible distortions. Goodfellow et al.[8] proposed the fast gradient sign method (FGSM), which perturbs a target sample to its gradient direction, to increase its loss, and also use the generated samples to train the model for improved robustness. Follow-up works [9, 19–21] proposed iterative variants of the gradient attack with improved adversarial learning frameworks. After these gradient-based attacks have become standard in evaluating the robustness of deep neural networks, many more defenses followed, but Athalye et al. [22] showed that many of them appear robust only because they mask out

the gradients, and proposed new types of attacks that circumvent gradient obfuscation. Recent works focus on the vulnerability of the latent representations, hypothesizing them as the main cause of the adversarial vulnerability of deep neural networks. TRADES [2] uses Kullback-Leibler divergence loss between a clean example and its adversarial counterpart to push the decision boundary, to obtain a more robust latent space. Ilyas et al. [23] showed the existence of imperceptible features that help with the prediction of clean examples but are vulnerable to adversarial attacks. On the other hand, instead of defending the adversarial attacks, guarantee the robustness become one of the solutions to the safe model. Li et al.[24], "randomized smoothing" technique has been empirically proposed as certified robustness. Then, Cohen et al. [25], prove the robustness guarantee of randomized smoothing in $\ell_2$ norm adversarial attack. Moreover, to improve the performance of randomized smoothing [26] directly attack the smoothed classifier. A common requirement of existing adversarial learning techniques is the availability of class labels, since they are essential in generating adversarial attacks. Recently, semi-supervised adversarial learning [16, 17] approaches have proposed to use unlabeled data and achieved large enhancement in adversarial robustness. Yet, they still require a portion of labeled data, and does not change the class-wise nature of the attack. Contrarily, in this work, we propose instance-wise adversarial attacks that do not require *any* class labels.

**Self-supervised learning**    As acquiring manual annotations on data could be costly, self-supervised learning, which generates supervised learning problems out of unlabeled data and solves for them, is gaining increasingly more popularity. The convention is to train the network to solve a manually-defined (pretext) task for representation learning, which will be later used for a specific supervised learning task (e.g., image classification). Predicting the relative location of the patches of images [11, 27, 28] has shown to be a successful pretext task, which opened the possibility of self-supervised learning. Gidaris et al. [10] propose to learn image features by training deep networks to recognize the 2D rotation angles, which largely outperforms previous self-supervised learning approaches. Corrupting the given images with gray-scaling [29] and random cropping [30], then restoring them to their original condition, has also shown to work well. Recently, leveraging the instance-level identity is becoming a popular paradigm for self-supervised learning due to its generality. Using the contrastive loss between two different views of the same images [15] or two different transformed images from one identity [12, 13, 31] have shown to be highly effective in learning the rich representations, which achieve comparable performance to fully-supervised models. Moreover, even with the labels, the contrastive loss leverage the performance of the model than using the cross-entropy loss [32].

**Self-supervised learning and adversarial robustness**    Recent works have shown that using un-labeled data could help the model to obtain more robust representations [16]. Moreover, [33] shows that a model trained with self-supervision improves the robustness. Using self-supervision signal in terms of perceptual loss also shows effective results in denoising the adversarial perturbation as purifier network [34]. Even finetuning the pretrained self-supervised learning helps the robustness [18], and self-supervised adversarial training coupled with the K-Nearest Neighbour classification improves the robustness of KNN [35]. However, to the best of our knowledge, none of these previous works explicitly target for adversarial robustness on unlabeled training. Contrarily, we propose a novel instance-wise attack, which leads the model to predict an incorrect instance for an instance-discrimination problem. This allows the trained model to obtain robustness that is on par or even better than supervised adversarial learning methods.

## 3    Adversarial Self-Supervised Learning with Instance-wise Attacks

We now describe how to obtain adversarial robustness in the representations *without* any class labels, using instance-wise attacks and adversarial self-supervised contrastive learning. Before describing ours, we first briefly describe supervised adversarial training and self-supervised contrastive learning.

**Adversarial robustness**    We start with the definition of adversarial attacks under supervised settings. Let us denote the dataset $\mathbb{D} = \{X, Y\}$, where $x \in X$ is training sample and $y \in Y$ is a corresponding label, and a supervised learning model $f_\theta : X \rightarrow Y$ where $\theta$ is parameters of the model. Given such a dataset and a model, *adversarial attacks* aim towards finding the worst-case examples nearby by searching for the perturbation, which maximizes the loss within a certain radius from the sample (e.g., norm balls). We can define such adversarial attacks as follows:

$$x^{i+1} = \Pi_{B(x,\epsilon)}(x^i + \alpha \texttt{sign}(\nabla_{x^i} \mathcal{L}_{\texttt{CE}}(\theta, x^i, y)) \tag{1}$$

---

**Algorithm 1** Robust Contrastive Learning (RoCL)

---

**Input:** Dataset $\mathbb{D}$, parameter of model $\theta$, model $f$, parameter of projector $\pi$, projector $g$, constant $\lambda$

    **for all** iter $\in$ number of training iteration **do**

        **for all** $x \in$ minibatch $B = \{x_1, \ldots, x_m\}$ **do**

            Generate adversarial examples from transformed inputs       ▷ *instance-wise* attacks

            $t(x)^{i+1} = \Pi_{B(t(x),\epsilon)}(t(x)^i + \alpha \mathtt{sign}(\nabla_{t(x)^i} \mathcal{L}_{\mathtt{con},\theta,\pi}(t(x)^i, \{t'(x)\}, t(x)_{\mathtt{neg}})))$

        **end for**

        $\mathcal{L}_{\mathtt{total}} = \frac{1}{N} \sum_{k=1}^{N} [\mathcal{L}_{\mathtt{RoCL},\theta,\pi} + \lambda \mathcal{L}_{\mathtt{con},\theta,\pi}(t(x)_k^{adv}, \{t'(x)_k\}, \{t(x)_{\mathtt{neg}}\})]$     ▷ total loss

        Optimize the weight $\theta, \pi$ over $\mathcal{L}_{total}$

    **end for**

---

where $B(x, \epsilon)$ is the $\ell_\infty$ norm-ball around $x$ with radius $\epsilon$, and $\Pi$ is the projection function for norm-ball. The $\alpha$ is the step size of the attacks and $\mathtt{sign}(\cdot)$ returns the sign of the vector. Further, $\mathcal{L}_{\mathtt{CE}}$ is the cross-entropy loss for supervised training, and $i$ is the number of attack iterations. This formulation generalizes across different types of gradient attacks. For example, Projected Gradient Descent (PGD) [9] starts from a random point within the $x \pm \epsilon$ and perform $i$ gradient steps, to obtain an attack $x^{i+1}$.

The simplest and most straightforward way to defend against such adversarial attacks is to minimize the loss of adversarial examples, which is often called *adversarial learning*. The adversarial learning framework proposed by Madry et al.[9] solve the following non-convex outer minimization problem and non-convex inner maximization problem where $\delta$ is the perturbation of the adversarial images, and $x + \delta$ is an adversarial example $x^{adv}$, as follow:

$$\underset{\theta}{\mathrm{argmin}} \, \mathbb{E}_{(x,y) \sim \mathbb{D}} \big[ \underset{\delta \in B(x,\epsilon)}{\max} \mathcal{L}_{\mathtt{CE}}(\theta, x + \delta, y) \big] \tag{2}$$

In standard adversarial learning framework, including PGD [9], TRADES [2], and many others, generating such adversarial attacks require to have a class label $y \in Y$. Thus, conventional adversarial attacks are inapplicable to unlabeled data.

**Self-supervised contrastive learning** The self-supervised contrastive learning framework [12, 13] aims to maximize the agreement between different augmentations of the same instance in the learned latent space while minimizing the agreement between different instances. Let us define some notions and briefly recap the SimCLR. To project the image into a latent space, SimCLR uses an encoder $f_\theta(\cdot)$ network followed by a projector, which is a two-layer multi-layer perceptron (MLP) $g_\pi(\cdot)$ that projects the features into latent vector z. SimCLR uses a stochastic data augmentation $t$, randomly selected from the family of augmentations $\mathcal{T}$, including random cropping, random flip, random color distortion, and random grey scale. Applying any two transformations, $t, t' \sim \mathcal{T}$, will yield two samples denoted $t(x)$ and $t'(x)$, that are different in appearance but retains the instance-level identity of the sample. We define $t(x)$'s positive set as $\{x_{\mathtt{pos}}\} = t'(x)$ from the same original sample $x$, while the negative set $\{x_{\mathtt{neg}}\}$ as the set of pairs containing the other instances $x'$. Then, the contrastive loss function $\mathcal{L}_{\mathtt{con}}$ can be defined as follows:

$$\mathcal{L}_{\mathtt{con},\theta,\pi}(x, \{x_{\mathtt{pos}}\}, \{x_{\mathtt{neg}}\})$$
$$:= -\log \frac{\sum_{\{z_{\mathtt{pos}}\}} \exp(\mathrm{sim}(z, \{z_{\mathtt{pos}}\})/\tau)}{\sum_{\{z_{\mathtt{pos}}\}} \exp(\mathrm{sim}(z, \{z_{\mathtt{pos}}\})/\tau) + \sum_{\{z_{\mathtt{neg}}\}} \exp(\mathrm{sim}(z, \{z_{\mathtt{neg}}\})/\tau)}, \tag{3}$$

where z, $\{z_{\mathtt{pos}}\}$, and $\{z_{\mathtt{neg}}\}$ are corresponding 128-dimensional latent vectors obtained by the encoder and projector $z = g_\pi(f_\theta(x))$, $\{x_{\mathtt{pos}}\}$, and $\{x_{\mathtt{neg}}\}$, respectively. The standard contrastive learning only contains a single sample in the positive set $\{x_{\mathtt{pos}}\}$, which is $t(x)$. The $\mathrm{sim}(u, v) = u^T v / \|u\|\|v\|$ denote cosine similarity between two vectors and $\tau$ is a temperature parameter.

We show that standard contrastive learning, such as SimCLR, is vulnerable to the adversarial attacks as shown in Table 1. To achieve robustness with such self-supervised contrastive learning frameworks, we need a way to adversarially train them, which we will describe in the next subsection.

### 3.1 Adversarial Self-supervised Contrative Learning

We now introduce a simple yet novel and effective approach to adversarially train a self-supervised learning model, using unlabeled data, which we coin as *robust contrastive learning (RoCL)*. RoCL

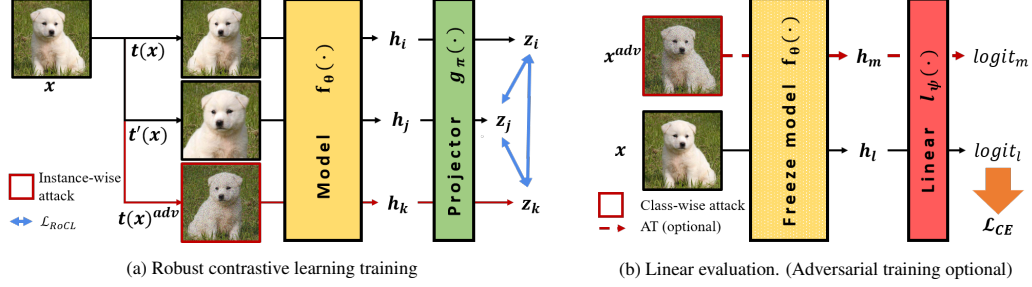

| (a) Robust contrastive learning training | (b) Linear evaluation. (Adversarial training optional) |

Figure 2: **Adversarial training and evaluation steps for RoCL.** During adversarial training, we maximize the similarity between two differently transformed examples $\{t(x), t'(x)\}$ and their adversarial perturbations $t(x)^{adv}$. After the model is fully trained to obtain robustness, then we evaluate the model on the target classification task by using linear model instead of projector. Here, we could either train the linear classifier only on clean examples, or adversarially train it with class-adversarial examples.

is trained without a class label by using *instance-wise* attacks, which makes the model confuse the instance-level identity of a given sample. Then, we use a contrastive learning framework to maximize the similarity between a transformed example and the instance-wise adversarial example of another transformed example. Algorithm 1 summarizes our robust contrastive learning framework.

**Instance-wise adversarial attacks**   Since class-wise adversarial attacks for existing approaches are inapplicable to the unlabeled case we target, we propose a novel *instance-wise* attack. Specifically, given a sample of an input instance, we generate a perturbation to fool the model by confusing its instance-level *identity*; such that it mistakes it as an another sample. This is done by generating a perturbation that maximizes the self-supervised contrastive loss for discriminating between the instances, as follows:

$$t(x)^{i+1} = \Pi_{B(t(x),\epsilon)}(t(x)^i + \alpha \texttt{sign}(\nabla_{t(x)^i} \mathcal{L}_{\texttt{con},\theta,\pi}(t(x)^i, \{t'(x)\}, \{t(x)_{\texttt{neg}}\}))) \quad (4)$$

where $t(x)$ and $t'(x)$ are transformed images with stochastic data augmentations $t, t' \sim \mathcal{T}$, and $\{t(x)_{\texttt{neg}}\}$ are the negative instances for $t(x)$, which are examples of other samples $x'$.

**Robust Contrastive Learning (RoCL)**   We now present a framework to learn robust representation via self-supervised contrastive learning. The adversarial learning objective for an instance-wise attack, following the min-max formulation of [9] could be given as follows:

$$\underset{\theta,\pi}{\arg\min} \, \mathbb{E}_{(x)\sim\mathbb{D}}[\underset{\delta\in B(t(x),\epsilon)}{\max} \mathcal{L}_{\texttt{con},\theta,\pi}(t(x) + \delta, \{t'(x)\}, \{t(x)_{\texttt{neg}}\})] \quad (5)$$

where $t(x) + \delta$ is the adversarial image $t(x)^{adv}$ generated by *instance-wise* attacks (eq. 4). Note that we generate the adversarial example of $x$ using a stochastically transformed image $t(x)$, rather than the original image $x$, which will allow us to generate diverse attack samples. This adversarial learning framework is essentially the same as the supervised adversarial learning framework, except that we train the model to be robust against m-way instance-wise adversarial attacks. Note that the proposed regularization can be interpreted as a denoiser. Since the contrastive objective maximize the similarity between clean samples: $t(x), t'(x)$, and generated adversarial example, $t(x)^{adv}$.

We generate label-free adversarial examples using instance-wise adversarial attacks in eq. 4. Then we use the contrastive learning objective to maximize the similarity between clean examples and their instance-wise perturbation. This is done using a simple modification of the contrastive learning objective in eq. 3, by using the *instance-wise* adversarial examples as additional elements in the positive set. Then we can formulate our *Robust Contrastive Learning* objective as follow:

$$\begin{aligned} \mathcal{L}_{\texttt{RoCL},\theta,\pi} &:= \mathcal{L}_{\texttt{con},\theta,\pi}(t(x), \{t'(x), t(x)^{adv}\}, \{t(x)_{\texttt{neg}}\}) \\ \mathcal{L}_{\texttt{total}} &:= \mathcal{L}_{\texttt{RoCL},\theta,\pi} + \lambda \mathcal{L}_{\texttt{con},\theta,\pi}(t(x)^{adv}, \{t'(x)\}, \{t(x)_{\texttt{neg}}\}) \end{aligned} \quad (6)$$

where $t(x)^{adv}$ are the adversarial perturbation of an augmented sample $t(x)$, $t'(x)$ is another stochastic augmentation, and $\lambda$ is a regularization parameter. The $\{z_{\texttt{pos}}\}$, which is the set of positive samples in the latent feature space, is compose of $z'$ and $z^{adv}$ which are latent vectors of $t'(x)$ and $t(x)^{adv}$ respectively. The $\{z_{\texttt{neg}}\}$ is the set of latent vectors for negative samples in $\{t(x)_{\texttt{neg}}\}$.

**Linear evaluation of RoCL**  With RoCL, we can adversarially train the model without any class labels (Figure 2(a)). Yet, since the model is trained for instance-wise classification, it cannot be directly used for class-level classification. Thus, existing self-supervised learning models leverage *linear evaluation* [12, 29, 36, 37], which learns a linear layer $l_\psi(\cdot)$ on top of the fixed $f_\theta(\cdot)$ embedding layer (Figure 2(b)) with clean examples. While RoCL achieves impressive robustness with this standard evaluation (Table 1), to properly evaluate the robustness against a specific type of attack, we propose a new evaluation protocol which we refer to as *robust-linear evaluation (r-LE)*. r-LE trains a *linear* classifier with class-level adversarial examples of specific attack (e.g. $\ell_\infty$) with the fixed encoder as follows:

$$\underset{\psi}{\arg\min}\, \mathbb{E}_{(x,y)\sim\mathbb{D}}[\max_{\delta\in B(x,\epsilon)} \mathcal{L}_{\text{CE}}(\psi, x + \delta, y)] \tag{7}$$

where $\mathcal{L}_{\text{CE}}$ is the cross-entropy that only optimize parameters of linear model $\psi$. While we propose r-LE as an evaluation measure, it could be also used as an efficient means of obtaining an adversarially robust network using network pretrained using self-supervised learning.

**Transformation smoothed inference**  We further propose a simple inference method for robust representation. Previous works [26, 25] proposed *smoothed classifiers*, which obtain smooth decision boundaries for the final classifier by taking an expectation over classifiers with Gaussian-noise perturbed samples. This method aims to fix the problem with the sharp classifiers, which may result in misclassification of the points even with small perturbations. Similarly, we observe that our objective enforces to assemble all differently transformed images into the adjacent area, and propose a *transformation smoothed classifier* to obtain a smooth classifier for RoCL, which predicts the class $c$ by calculating expectation $\mathbb{E}$ over the transformation $t \sim \mathcal{T}$ for a given input $x$ as follows:

$$S(x) = \underset{c\in Y}{\arg\max}\, \mathbb{E}_{t\sim\mathcal{T}}(l_c(f(t(x))) = c) \tag{8}$$

where $l_c(.)$ is the logit value of the class. We approximate the expectation over the transformation by multiple sampling the random transformation $t$ and aggregating the penultimate feature $f(t(x))$.

## 4    Experimental Results

We now validate RoCL on benchmark datasets against existing adversarial learning methods. Specifically, we report the results of our model against white-box and black-box attacks and in the transfer learning scenario in Section 4.1, and conduct an ablation study to verify the efficacy of individual component of RoCL in Section 4.2.

**Experimental setup**  For every experiments in the main text, we use ResNet18 or ResNet50 [38] trained on CIFAR-10 [39]. For all baselines and our method, we train with $\ell_\infty$ attacks with the same attack strength of $\epsilon = 8/255$. All ablation studies are conducted with ResNet18 trained on CIFAR-10, with the attack strength of $\epsilon = 8/255$. Regarding the additional results on CIFAR-100 and details of the optimization & evaluation, please see the Appendix A, and C. The code to reproduce the experimental results is available at `https://github.com/Kim-Minseon/RoCL`.

### 4.1   Main Results

We first report the results of baselines and our models against white-box attacks with linear evaluation, robust linear evaluation and finetuning in Table 1. We also report the results against black-box attacks in Table 2, where adversarial samples are generated by AT, TRADES, RoCL with the PGD attack, and RoCL model with the instance-wise attack. Then, we demonstrate the efficacy of the transformation smoothed classifier in Table 3. We further report the results of transfer learning, where we transfer the learned networks from from CIFAR-10 to CIFAR-100, and CIFAR-100 to CIFAR-10 in Table 4.

**Results on white box attacks**  To our knowledge, our *RoCL* is the first attempt to achieve robustness in a fully self-supervised learning setting, since existing approaches used self-supervised learning as a pretraining step before supervised adversarial training. Therefore, we analyze the robustness of representation of RoCL which is acquire during the training only with linear evaluation including robust linear evaluation. Also, we discover that RoCL is also robust against unseen attacks. Lastly, we demonstrate the results of finetuning the RoCL.

Table 1: Experimental results with white box attacks on ResNet18 and ResNet50 trained on the CIFAR-10 . r-LE denotes robust linear evaluation, and SCL is the supervised contrastive learning [32] which uses the labels in the contrastive loss. Baselines with * are the models with our data augmentation applied during training. AT denotes the supervised adversarial training[9], and SS denotes the self-supervised loss. Rot+pretrained is the model [18] which finetunes the network trained with rotation-prediction self-supervised learning. For a fair comparison, we report the single self-supervised model pretrained version with the ResNet50-v2 model. $^+$ is the reported performance of [18]. $A_{nat}$ is the accuracy of clean images. All models are trained with $\ell_\infty$; thus the $\ell_\infty$ is the *seen* adversarial attack and $\ell_2$, and $\ell_1$ attacks are *unseen*.

| Train type | Method | ResNet18 | | | | | | | ResNet50 | | | | | | |
| | | | seen | | unseen | | | | | seen | | unseen | | | |
| | | | $\ell_\infty$ | | $\ell_2$ | | $\ell_1$ | | | $\ell_\infty$ | | $\ell_2$ | | $\ell_1$ | |
| | | $A_{nat}$ | $\epsilon$ 8/255 | 16/255 | 0.25 | 0.5 | 7.84 | 12 | $A_{nat}$ | $\epsilon$ 8/255 | 16/255 | 0.25 | 0.5 | 7.84 | 12 |
|---|---|---|---|---|---|---|---|---|---|---|---|---|---|---|---|
| Supervised | $\mathcal{L}_{CE}$ | 92.82 | 0.00 | 0.00 | 20.77 | 12.96 | 28.47 | 15.56 | 93.12 | 0.00 | 0.00 | 13.42 | 3.44 | 28.78 | 13.98 |
| | AT[9] | 81.63 | 44.50 | 14.47 | 72.26 | **59.26** | **66.74** | 55.74 | 84.03 | 46.76 | 17.63 | 72.98 | 58.78 | 65.28 | 52.45 |
| | TRADES[2] | 77.03 | **48.01** | **22.55** | 68.07 | 57.93 | 62.93 | 53.79 | 82.10 | 53.49 | 25.18 | **73.01** | **61.94** | 65.48 | 54.52 |
| | TRADES*[2] | 73.26 | 42.71 | 17.71 | 65.25 | 56.13 | 62.89 | **55.95** | 75.65 | 46.20 | 20.96 | 67.02 | 57.12 | 62.46 | 55.09 |
| | SCL[32] | **94.05** | 0.08 | 0.00 | 22.17 | 10.29 | 38.87 | 22.58 | **95.02** | 0.00 | 0.00 | 16.72 | 1.68 | 39.44 | 22.59 |
| Self -supervised | SimCLR[12] | **91.25** | 0.63 | 0.08 | 15.3 | 2.08 | 41.49 | 25.76 | **92.69** | 0.07 | 0.00 | 25.13 | 3.85 | 50.17 | 31.63 |
| | **RoCL** | 83.71 | 40.27 | 9.55 | 66.39 | 63.82 | **79.21** | **76.17** | 85.99 | 43.56 | 11.38 | **70.87** | **67.59** | **82.65** | **80.02** |
| | **RoCL+rLE** | 80.43 | **47.69** | **15.53** | 68.30 | 66.19 | 77.31 | 75.05 | 80.79 | **45.33** | **16.85** | 67.14 | 64.61 | 77.54 | 75.76 |
| Self -supervised +finetune | Rot. Pretrained[18] | - | - | - | - | - | - | - | 85.66$^+$ | 50.40$^+$ | - | - | - | - | - |
| | **RoCL+AT** | 80.26 | 40.77 | **22.83** | 68.64 | 56.25 | 65.16 | 56.07 | 82.72 | 50.60 | 18.83 | 72.12 | **70.03** | 81.02 | 79.22 |
| | **RoCL+TRADES** | 84.55 | 43.85 | 14.29 | **73.01** | 60.03 | 68.25 | 58.04 | 85.41 | 45.68 | 21.21 | 74.06 | 59.60 | 65.37 | 53.54 |
| | **RoCL+AT+SS** | 91.34 | **49.66** | 14.44 | 70.75 | **61.55** | **83.08** | **81.18** | 84.67 | **52.44** | 19.53 | **76.61** | 66.38 | 72.76 | 64.56 |

Table 2: Performance of RoCL against black box attacks on the CIFAR-10 dataset. Each column denotes the black box model used to generate the $\ell_\infty$ adversarial examples with $\epsilon = 8/255$ and 16/255, respectively. We generate PGD attack examples (PGD) and instance-wise attack examples (*inst.*) from RoCL with linear layer and a projector, respectively. Each row shows the performance of the target model trained with $\ell_\infty$.

| Target \ Source | ResNet18 | | | | | | | |
| | 8/255 | | | | 16/255 | | | |
| | AT | TRADES | RoCL(PGD) | RoCL(*inst.*) | AT | TRADES | RoCL(PGD) | RoCL(*inst.*) |
|---|---|---|---|---|---|---|---|---|
| AT [9] | - | **77.48** | 69.83 | 47.25 | - | **63.87** | 48.99 | 47.42 |
| TRADES [2] | 60.73 | - | 64.81 | 46.22 | 41.87 | - | 48.07 | 45.73 |
| RoCL | **66.76** | 77.33 | - | - | **41.97** | 62.98 | - | - |

We first compare RoCL against SimCLR[12], which is a vanilla self-supervised contrastive learning model. The result shows that SimCLR is extremely vulnerable to adversarial attacks. However, RoCL achieves high robust accuracy (40.27) against the target $\ell_\infty$ attacks. This is an impressive result, which demonstrates that it is possible to train adversarially robust models without any labeled data. Moreover, RoCL+rLE outperform supervised adversarial training by Madry et al. [9] and obtains comparable performance to TRADES [2]. Note that while we used the same number of instances in this experiment, in practice, we can use any number of unlabeled data available to train the model, which may lead to larger performance gains. To show that this result is not due to the effect of using augmented samples for self-supervised learning, we applied the same set of augmentations for TRADES (TRADES*), but it obtains worse performance over the original TRADES.

Moreover, RoCL obtains significantly higher robustness over the supervised adversarial learning approaches against *unseen* types of attacks, except for $\ell_1$ attack with small perturbation, and much higher clean accuracy (See the results on $\ell_2$, $\ell_1$ attacks in Table 1). This makes RoCL more appealing over baselines in practice, and suggests that our approach to enforce a consistent identity over diverse perturbations of a single sample in the latent representation space is a more fundamental solution to ensure robustness against general types of attacks. This point is made more clear in the comparison of RoCL against RoCL with robust linear evaluation (RoCL+rLE), which trains the linear classifier with class-wiser adversaries. RoCL+rLE improves the robustness against the target $\ell_\infty$ attacks, but degenerates robustness on unseen types of attacks ($\ell_1$).

Existing works [40, 18] have shown that finetuning the supervised or self-supervised pretrained networks with adversarial training improves robustness. This is also confirmed with our results in Table 1, which show that the models fine-tuned with our method obtain even better robustness and higher clean accuracy over models trained from scratch. We observe that using self-supervised loss (SS loss eq. 3) during adversarial finetuning further improves robustness (RoCL + AT + SS). Moreover, our method outperforms Chen et al. [18], which uses self-supervised learning only for model pretraining, before supervised adversarial training.

Table 3: Experimental results of transformation smoothed classifier. We test the RoCL + transformation smoothed classifier against clean images, black box attack images which is generated from AT attack and expectation of transformation attack (EoT).

| | Clean | Blackbox (AT) | | EoT |
|---|---|---|---|---|
| | - | 8/255 | 16/255 | 8/255 |
| RoCL | 84.11 (+0.40) | 66.86 (+0.10) | 42.50 (+0.43) | 33.24 - |

Table 4: Results of transfer learning across the CIFAR-10 and CIFAR-100 datasets with ResNet18. We compare against adversarial transfer learning results from [41], with a larger WRN 32-10 [42] architecture. $^{+}$ is the reported performance from [41].

| source | target | Method | $A_{nat}$ | $\ell_\infty$ |
|---|---|---|---|---|
| CIFAR-100 | CIFAR-10 | Transfer$^{+}$[41] | 72.05 | 17.70 |
| | | **RoCL** | **73.93** | **18.62** |
| CIFAR-10 | CIFAR-100 | Transfer$^{+}$[41] | 41.59 | 11.63 |
| | | **RoCL** | **45.84** | **15.33** |

Table 5: Performance with different target images for generating instance-wise attacks.

| | $A_{nat}$ | 8/255 | 16/255 |
|---|---|---|---|
| original $x$ | **87.96** | 36.6 | **11.78** |
| $t'(x)$ | 83.71 | **40.27** | 9.55 |

Table 7: Performance of RoCLs with different attack loss types. The original RoCL maximizes the contrastive loss (Contrastive) to generate instance-wise attacks. We observe that Other types of losses, such as mean square error (MSE), cosine similarity, Manhattan distance (MD) are less effective.

| $\mathcal{L}_{\theta,\pi}$ | $A_{nat}$ | 8/255 | 16/255 |
|---|---|---|---|
| Contrastive | 83.71 | **40.27** | **9.55** |
| MSE | **88.35** | 40.12 | 7.88 |
| Cosine similarity | 73.49 | 9.30 | 0.06 |
| MD | 84.40 | 21.05 | 1.65 |

Table 6: Experimental results of RoCL against $\ell_\infty$ attack with different number of steps.

| | 20 | 40 | 100 |
|---|---|---|---|
| RoCL | 40.27 | 39.80 | 39.74 |

**Results on black box attacks**   We also validate our models against black-box attacks. We generate adversarial examples using the AT, TRADES, and RoCL, perform black-box attacks across the methods. As shown in Table 2, our model is superior to TRADES [2] against AT black box attacks, and achieves comparable performance to AT [9] against TRADES black box attack samples. We also validate RoCL's robustness by generating adversarial samples using our model and use them to attack AT and TRADES. We also generate black-box adversarial examples with RoCL by attacking the RoCL with a linear layer using the PGD attack (RoCL (PGD)), and the RoCL with a projector using the instance-wise attack (RoCL (*inst.*)). The low robustness of attacked models (AT, TRADES) shows that attacks with RoCL are strong. Specifically, RoCL with the PGD attack is stronger than TRADES attacks on AT, and RoCL with the instance-wise attacks is significantly stronger over both AT and TRADES black box attacks.

**Transformation smoothed classifier**   Transformation smoothed classifier can enhance the model accuracy not only on the black-box adversarial examples, but also on clean examples (Table 3). Intuitively, since we enforce differently transformed samples of the same instance to have a consistent identity, they will be embedded in nearby places in the latent representation space.  Therefore, we can calculate the transformation ball around the samples, that is similar to Gaussian ball in [25]. Accordingly, RoCL obtains a smoother classifier and acquires larger gains in both black-box robustness and clean accuracy (Table 3). As shown in Figure 3(d), as the number of samples ($t \sim \mathcal{T}$) increases, the model becomes increasingly more robust. We also test the transformation smoothed classifier with expectation of transformation (EoT) attack [22], which is a white box attack against models with test-time randomness. We found that although transformation smoothed classifier suffers from loss of robust accuracy with EoT attacks, it is still reasonably robust (Table 3). We provide the detailed settings of transformation smoothed classifier experiments in Section A of the Appendix.

**Transfer learning**   Another advantage of our unsupervised adversarial learning, is that the learned representations can be easily transferred to diverse target tasks. We demonstrate the effectiveness of our works on transfer learning in Table 4, against the fully supervised adversarial transfer learning [41] with larger networks. Surprisingly, our model achieves even better accuracy and robustness in both cases (CIFAR-10→CIFAR-100 and CIFAR-100→CIFAR-10) without any other additional losses. The detailed settings for the transfer learning experiments are given in Section B of the Appendix .

### 4.2   Ablation studies

**Effect of target images to generate attacks**   When generating instance-wise attacks, we can either attack the original $x$ or the transformed instance $t'(x)$. The comparative study in Table 5 shows that our RoCL achieves high clean accuracy and robustness regardless of the target examples we select for instance-wise perturbation. This is because the our method aims at preserving the instance-level

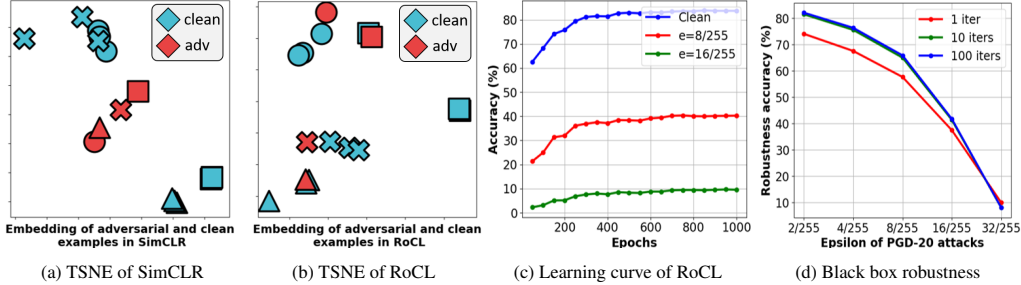

|  (a) TSNE of SimCLR | (b) TSNE of RoCL | (c) Learning curve of RoCL | (d) Black box robustness |

Figure 3: (a,b) Visualizations of the embedding of *instance-wise* adversarial examples and clean examples for SimCLR and RoCL after training. (c) The learning curve of ResNet18 RoCL. (d) The transformation smoothed classifier performance on AT's black box attack over transformation iterations against different attack budgets.

identity regardless of the transformations applied to an instance. Therefore, our methods achieves consistent performance with any target instances that have the same identity.

**Effect of attack loss type**    For instance-wise attacks, we can consider various losses to maximize the distance of adversarial samples from the target samples. We compare four different distance functions, namely mean square error (MSE), cosine similarity, Manhattan distance (MD), and contrastive loss. Table 7 shows that the contrastive loss is the most effective among all losses we considered.

**Effect of the number of PGD attack iterations**    We further validate the robustness of RoCL under larger iteration steps of the PGD attack. Table 6 shows that RoCL remains robust with any number of PGD iterations (e.g., 39.74% under 100 iteration steps).

**Visualizations of *instance-wise* attacks**    We further examine and visualize the samples generated with our instance-wise attacks on SimCLR in Figure 3(a)). The visualization of the samples in the latent embedding space shows that our attacks generate confusing samples (denoted with red markers) that are far apart from the original instances (denoted with blue markers) with the same identities. However, after we train the model with RoCL (Figure 3(b)), the instance-wise adversarial examples are pushed toward the samples with the same instance-level identity.

## 5   Conclusion

In this paper, we tackled a novel problem of learning robust representations without any class labels. We first proposed a *instance-wise attack* to make the model confuse the instance-level identity of a given sample. Then, we proposed a *robust contrastive learning* framework to suppress their adversarial vulnerability by maximizing the similarity between a transformed sample and its instance-wise adversary. Furthermore, we demonstrate an effective transformation smoothed classifier which boosts our performance during the test inference. We validated our method on multiple benchmarks with different neural architectures, on which it obtained comparable robustness to the supervised baselines on the targeted attack without any labels. Notably, RoCL obtained significantly better clean accuracy and better robustness against black box, unseen attacks, and transfer learning, which makes it more appealing as a general defense mechanism. We believe that our work opened a door to more interesting follow-up works on *unsupervised adversarial learning*, which we believe is a more fundamental solution to achieving adversarial robustness with deep neural networks.

## Broader Impact

Achieving adversarial robustness against malicious attacks with deep neural networks, is a fundamental topic of deep learning research that has not yet been fully solved. Until now, supervised adversarial training, which perturbs the examples such that the target deep network makes incorrect predictions, has been a dominant paradigm in adversarial learning of deep neural networks. However, supervised adversarial learning suffers from lack of generalization to unseen types of attacks, or unseen datasets, as well as suffers from loss of accuracy on clean examples, and thus is not a fundamental, nor practical solution to the problem. Our adversarial self-supervised learning is a research direction that delved into the vulnerability of deep networks in the intrinsic representation space, which we believe is the root cause of fragility of existing deep neural networks, and we hope that more research is conducted in the similar directions.

## Acknowledgements

This work was supported by Institute of Information & communications Technology Planning & Evaluation (IITP) grant funded by the Korea government (MSIT) (No.2020-0-00153) and Samsung Research Funding Center of Samsung Electronics (No. SRFC-IT1502-51). We thank Sihyun Yu, Seanie Lee, and Hayeon Lee for providing helpful feedbacks and suggestions in preparing an earlier version of the manuscript. We also thank the anonymous reviewers for their insightful comments and suggestions.

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
