[Supplementary Material · Final_RoCL_removed.pdf]

# Appendix
## Adversarial Self-Supervised Contrastive Learning

**Organization**   Appendix is organized as follows. In section A, we describe the experimental details, including the descriptions of the datasets and the evaluation process. We then provide an algorithm which summarizes our RoCL in section B. Then, we further report the RoCL results on both CIFAR-10 and CIFAR-100 against PGD attacks and CW attacks in Section C. Finally, perform ablation studies of our RoCL in section D.

## A   Experimental Setup

### A.1   Training detail and dataset

**Training details** We use ResNet18 and ResNet50 [38] as the base encoder network $f_\theta$ and 2-layer multi-layer perceptron with 128 embedding dimension as the projection head $g_\pi$. All models are trained by minimizing the final loss $\mathcal{L}_{\texttt{total}}$ with a temperature of $\tau = 0.5$. We set the regularization parameter to $\lambda = 1/256$. For the inner maximization step of RoCL i.e., instance-wise attack, we set the perturbation $\epsilon = 0.0314$ and step size $\alpha = 0.007$ under $\ell_\infty$ bound, with the number of inner maximize iteration as $K = 7$. For the rest, we follow the similar optimization step of SimCLR [12]. For optimization, we train RoCL with 1,000 epoch under LARS optimizer [43] with weight decay of $1e-6$ and momentum with 0.9. For the learning rate scheduling, we use linear warmup [44] for early 10 epochs until learning rate of 1.0 and decay with cosine decay schedule without a restart [45]. We use batch size of 512 for RoCL (we found out that batch size of 512 was sufficient for CIFAR-10 and CIFAR-100). Furthermore, we use global batch normalization (BN) [46], which shares the BN mean & variance in distributed training over the GPUs.

**Data augmentation details**  We use SimCLR augmentations: Inception crop [47], horizontal flip, color jitter, and grayscale for random augmentations. The detailed description of the augmentations are as follows. *Inception crop*: Randomly crops the area of the original image with uniform distribution 0.08 to 1.0. After the crop, cropped image are resized to the original image size. *Horizontal flip*: Flips the image horizontally with 50% of probability. *Color jitter*: Change the hue, brightness, and saturation of the image. We transform the RGB (red, green, blue) channeled image into an HSV (hue, saturation, value) channeled image format and add noise to the HSV channels. We randomly apply color jitter transformation with 80% of probability. *Grayscale*: Convert into a gray scale image. We randomly apply the grayscale transformation with 20% of probability.

**Dataset details** For RoCL training, we use CIFAR-10 [48] and CIFAR-100 [48]. CIFAR-10 and CIFAR-100 consist of 50,000 training and 10,000 test images with 10 and 100 image classes, respectively.

### A.2   Evaluation

**Linear evaluation setup**   In the linear evaluation phase, we train the linear layer $l_\psi$ on the top of the frozen encoder $f_\theta$. We train the linear layer for 150 epochs with the learning rate of 0.1. The learning rate is dropped by a factor of 10 at 30, 50, 100 epoch of the training progress. We use stochastic gradient descent (SGD) optimizer with a momentum of 0.9, weight decay of $5e$-4, and train the linear layer with the cross-entropy (CE) loss.

**Robust linear evaluation setup**   For robust linear evaluation, we train the linear layer $l_\psi$ on the top of the frozen encoder $f_\theta$, as done with linear evaluation. We train the linear layer for 150 epochs with an learning rate of 0.02. The learning rate scheduling and the optimizer setup is the same with the setup for linear evaluation. We use the project gradient descent (PGD) attack to generate class-wise adversarial examples. We perform $\ell_\infty$ attack with epsilon $\epsilon = 0.0314$ and the step size $\alpha = 0.007$ for 10 steps.

**Robustness evaluation setup**   For evaluation of adversarial robustness, we use white-box project gradient descent (PGD) attack. We evaluate under PGD attacks with 20, 40, 100 steps. We set $\ell_\infty, \ell_2, \ell_1$ attacks with $\epsilon = 0.0314, 0.072$ for $\ell_\infty$, $\epsilon = 0.25, 0.5$ for $\ell_2$, and $\epsilon = 7.84, 12$ for $\ell_1$ for testing CIFAR 10 and CIFAR 100.

### A.3   Transformation smoothed classifier setup

In the transformation smoothed classifer, we used same data augmentation that is used in training phase A.1. The probability is also same with training phase, yet we used fixed sized inception crop with 0.54 scale. For Table 3, we used 30 times iteration for all tests. For Figure 3(d) we differ the transformation iterations to 1, 10, and 100.

For the expectation of transformation (EoT), we evaluate under the perturbation $\epsilon = 0.0314$ and step size $\alpha = 0.00314$ under $\ell_\infty$ bound, with the number of inner maximize step iteration as $K = 20$.

## A.4 Transfer learning setup

We first briefly describe robust transfer learning and our experiments in its experimental setting. Shafahi et al. [41] suggest that an adversarially trained model can be transferred to another model to improve upon its robustness. They used modified WRN 32-10 to train the fully supervised adversarial model. Moreover, they initialize the student network with an adversarially trained teacher network and utilize the distillation loss and cross-entropy loss to train the student network's linear layer on the top of the encoder layer. We follow the experimental settings of Shafahi et al. [41], and train only the linear layer with cross-entropy loss. However, we did not use the distillation loss in order to evaluate the robustness of the encoder trained with our RoCL only (ResNet18). We train the linear model with CIFAR-100 on top of the frozen encoder, which is trained on CIFAR-10. We also train the linear layer with CIFAR-10 on top of the frozen encoder, which is trained on CIFAR-100. We train the linear layer for 100 epochs with a learning rate of 0.2. We use stochastic gradient descent (SGD) for optimization.

## A.5 Training efficiency of RoCL

**Training efficiency of RoCL** RoCL takes about 41.7 hours to train 1000 epochs with two RTX 2080 GPUs. Moreover, ours acquires sufficiently high clean accuracy and robustness even after 500 epochs (Figure 3(c)).

**Comparison to Semi-supervised learning in required dataset** Recently, semi-supervised learning[16, 17] have been shown to largely enhance the adversarial robustness of deep networks, by exploiting unlabeled data. However, they eventually require labeled data, to generate pseudo-labels on the unlabeled samples, and to generate class-wise adversaries. Also, they assume the availability of a larger dataset to improve robustness on the target dataset and require extremely large computation resources.

# B  Algorithm of RoCL

We present the algorithm for RoCL in Algorithm 2. During training, we generate the instance-wise adversarial examples using contrastive loss and then train the model using two differently transformed images and their instance-wise adversarial perturbations. We also include a regularization term that is defined as a contrastive loss between the adversarial examples and clean transformed examples.

---

**Algorithm 2** Robust Contrastive Learning (RoCL)

---

**Input:** Dataset $\mathbb{D}$, parameter of model $\theta$, model $f$, parameter of projector $\pi$, projector $g$, constant $\lambda$
  **for all** iter $\in$ number of training iteration **do**
    **for all** $x \in$ minibatch $B = \{x_1, \dots, x_m\}$ **do**
      Generate adversarial examples from transformed inputs          $\triangleright$ *instance-wise* attacks
      $t(x)^{i+1} = \Pi_{B(t(x),\epsilon)}(t(x)^i + \alpha \mathtt{sign}(\nabla_{t(x)^i} \mathcal{L}_{\mathtt{con},\theta,\pi}(t(x)^i, \{t'(x)\}, t(x)_{\mathtt{neg}})))$
    **end for**
    $\mathcal{L}_{\mathtt{total}} = \frac{1}{N} \sum_{k=1}^{N}[\mathcal{L}_{\mathtt{RoCL},\theta,\pi} + \lambda \mathcal{L}_{\mathtt{con},\theta,\pi}(t(x)_k^{adv}, \{t'(x)_k\}, \{t(x)_{\mathtt{neg}}\})]$          $\triangleright$ total loss
    Optimize the weight $\theta$, $\pi$ over $\mathcal{L}_{total}$
  **end for**

---

# C  Results of CIFAR-10 and CIFAR-100

While we only report the performance of RoCL on CIFAR-10 in the main paper as the baselines we mainly compare against only experimented on this dataset, we further report the performance of RoCL on CIFAR-100 as well (Table 8) and performance against CW attacks [21] (Table 9). We observe that RoCL consistently achieves comparable performance to that of the supervised adversarial learning methods, even on the CIFAR-100 dataset. Moreover, when employing the robust linear evaluation, RoCL acquires better robustness over the standard linear evaluation. Finally, the transformation smoothed classifier further boosts the performance of RoCL on both datasets.

Table 8: Experimental results with white box attacks on ResNet18 trained on the CIFAR-10 and CIFAR-100 dataset. r-LE denotes robust linear evaluation. AT denotes the supervised adversarial training[9]. All models are trained with $\ell_\infty$; thus the $\ell_\infty$ is the *seen* adversarial attack and $\ell_2$, and $\ell_1$ attacks are *unseen*.

| Train type | Method | CIFAR10 | | | | | | | | CIFAR100 | | | | | | | |
|---|---|---|---|---|---|---|---|---|---|---|---|---|---|---|---|---|---|
| | | | seen | | unseen | | | | | | seen | | unseen | | | |
| | | | $\ell_\infty$ | | $\ell_2$ | | $\ell_1$ | | | | $\ell_\infty$ | | $\ell_2$ | | $\ell_1$ | |
| | | $A_{nat}$ | $\epsilon$ 8/255 | 16/255 | 0.25 | 0.5 | 7.84 | 12 | $A_{nat}$ | | $\epsilon$ 8/255 | 16/255 | 0.25 | 0.5 | 7.84 | 12 |
| Supervised | $\mathcal{L}_{\mathtt{CE}}$ | 92.82 | 0.00 | 0.00 | 20.77 | 12.96 | 28.47 | 15.56 | 71.35 | | 0.00 | 0.00 | 6.54 | 2.31 | 11.14 | 5.86 |
| | AT[9] | 81.63 | 44.50 | 14.47 | **72.26** | **59.26** | **66.74** | **55.74** | 53.97 | | **20.09** | **6.19** | 43.08 | 32.29 | 40.43 | 33.18 |
| | TRADES[2] | 77.03 | **48.01** | **22.55** | 68.07 | 57.93 | 62.93 | 53.79 | 56.63 | | 17.94 | 4.29 | **44.82** | **33.76** | **43.70** | **37.00** |
| Self -supervised | SimCLR[12] | 91.25 | 0.63 | 0.08 | 15.3 | 2.08 | 41.49 | 25.76 | **57.46** | | 0.04 | 0.02 | 6.58 | 0.7 | 19.27 | 12.1 |
| | **RoCL** | 83.71 | 40.27 | 9.55 | 66.39 | 63.82 | **79.21** | **76.17** | 56.13 | | 19.31 | 4.30 | 38.65 | 35.94 | **50.21** | **46.67** |
| | **RoCL + rLE** | 80.43 | **47.69** | **15.53** | **68.30** | **66.19** | 77.31 | 75.05 | 51.82 | | **26.27** | **8.94** | **41.59** | **39.86** | 49.00 | 46.91 |

Table 9: Experimental results with white box CW attacks [21] on ResNet18 trained on the CIFAR-10. r-LE denotes robust linear evaluation. All models are trained with $\ell_\infty$

| Train type | Method | CIFAR-10 | | CIFAR-100 | |
|---|---|---|---|---|---|
| | | $A_{nat}$ | CW | $A_{nat}$ | CW |
| Self -supervised | **RoCL** | 83.71 | 77.35 | 56.13 | 44.57 |
| | **RoCL+rLE** | 80.43 | 76.15 | 51.82 | 44.77 |

# D Ablation

In this section, we report the results of several ablation studies of our RoCL model. For all experiments, we train the backbone network with 500 epochs and train the linear layer with 100 epochs, which yield models with sufficiently high clean accuracy and robustness. We first examine the effects of the target image when generating the instance-wise adversarial examples. Along with instance-wise attacks, the regularization term in algorithm 1 can also affect the final performance of the model. To examine lambda's effect on the transformed images, we set lambda as $\lambda = 1/256$ for CIFAR-10 and CIFAR-100. We also examine the effects of lambda $\lambda$ on the CIFAR-10 dataset.

## D.1 Adversarial contrastive learning

We examine the effect of the transformation function on the instance-wise attack and the regularization. For each input instance $x$, we generated three transformed images $t(x), t'(x)$, and $t(x)^{adv}$ and use them as the positive set. The results in Table 10 demonstrate that using any transformed images from the same identity for instance-wise attacks is equally effective. In contrast, for regularization, using images transformed with a different transformation function from the one used to generate attack helps obtain improved clean accuracy and robustness.

**Instance-wise attack** To generate instance-wise attacks, we can decide which identity we will use for instance-wise attack. Since the original transformed image $t(x)$ and image transformed with another transformation $t'(x)$ have the same identity, we can use both of them in instance-wise attacks. To find the optimal perturbation that maximizes the contrastive loss between adversarial examples and same identity images, we vary $\mathbf{X}$ in the following equation:

$$t(x)^{i+1} = \Pi_{B(t(x),\epsilon)}(t(x)^i + \alpha \texttt{sign}(\nabla_{t(x)^i} \mathcal{L}_{\texttt{con},\theta,\pi}(t(x)^i, \{\mathbf{X}\}, t(x)_{\texttt{neg}}))) \tag{9}$$

where $\mathbf{X}$ is either $t'(x)$ and $t(x)$.

**Regularization** To regularize the learning, we can calculate the contrastive loss between adversarial examples and clean samples with the same instance-level identity. We vary $\mathbf{Y}$ in the regularization term to examine which identity is the most effective, as follows:

$$\lambda \mathcal{L}_{\texttt{con},\theta,\pi}(t(x)_i^{adv}, \{\mathbf{Y}\}, \{t(x)_{\texttt{neg}}\}) \tag{10}$$

where $\mathbf{Y}$ can be $t'(x)$ and $t(x)$.

Table 10: Experimental results with white box attacks on ResNet18 trained on the CIFAR-10 and CIFAR-100 dataset. All models are trained with $\ell_\infty$.

| Method | instance-wise attack ($\mathbf{X}$) $t'(x)$ | $t(x)$ | regularization ($\mathbf{Y}$) $t'(x)$ | $t(x)$ | CIFAR-10 $A_{nat}$ | $\ell_\infty$ | CIFAR-100 $A_{nat}$ | $\ell_\infty$ |
|---|---|---|---|---|---|---|---|---|
| RoCL | ✓ | - | ✓ | - | **82.79** | **36.71** | **55.64** | **17.56** |
| | ✓ | - | - | ✓ | 81.47 | 29.97 | 53.84 | 14.18 |
| | - | ✓ | ✓ | - | 82.43 | 34.93 | 55.61 | 17.42 |
| | - | ✓ | - | ✓ | 81.96 | 30.99 | 53.76 | 14.74 |

## D.2  Lambda $\lambda$ and batch size $B$

We observe that $\lambda$, which controls the amount of regularization in the robust contrastive loss, and the batch size for calculating the contrastive loss, are two important hyperparameters for our robust contrastive learning framework. We examine the effect of two hyperparameters in Table 11, and Table 12. We observe that the optimal lambda $\lambda$ is different for each batch size $B$.

Table 11: lambda $\lambda$ ablation experimental results with white box attacks on ResNet18 trained on the CIFAR-10 dataset. All models are trained with $\ell_\infty$.

| CIFAR-10 | $\lambda$ | $A_{nat}$ | $\ell_\infty$ 8/255 | 16/255 |
|---|---|---|---|---|
| **RoCL** | 1/16 | 82.05 | 35.12 | 8.05 |
| | 1/32 | 82.25 | 36.02 | **8.68** |
| | 1/64 | **83.00** | 36.26 | 8.19 |
| | 1/128 | 82.79 | 36.71 | 8.34 |
| | 1/256 | 82.12 | **38.05** | 8.52 |
| | 1/512 | 82.68 | 37.24 | 8.53 |

Table 12: Ablation study of the batch size $B$, for the white box attacks on ResNet18 trained on the CIFAR-10 dataset. All models are trained with $\ell_\infty$ attacks.

| CIFAR-10 | $B$ | $\lambda$ | $A_{nat}$ | $\ell_\infty$ 8/255 | 16/255 |
|---|---|---|---|---|---|
| **RoCL** | 256 | 1/128 | 82.70 | 37.13 | 8.98 |
| | 256 | 1/256 | 82.90 | 36.86 | 8.89 |
| | 512 | 1/256 | 82.12 | 38.05 | 8.52 |
| | 1024 | 1/256 | 81.48 | 34.98 | 7.42 |