[Reviews · NeurIPS 2020]

Review 1

Summary and Contributions: This paper takes the first attempt to obtain robust models in an unsupervised manner. Specifically, this work is built upon the SimCLR framework, but additionally add adversarial examples as another positive instance during the contrastive learning process. Extensive results on CIFAR-10 are provided to demonstrate the effectiveness of the proposed method.

Strengths: (1) It is the first work that successfully shows we can learn robust models in an unsupervised manner. (2) The empirical results are pretty encouraging: (a) in terms of accuracy and robustness, unsupervised learned models can be on par with supervised learned models; (b) unsupervised learned models have much better performance on defending against black-box attacks and unseen attacks. (3) It is good that the authors provide some visualization to support that RoCL indeed learns a robust feature embedding.

Weaknesses: I have several major concerns on the presentations of this paper: (1) The proposed transformation smoothed inference may cause gradient obfuscation, therefore Expectation of Transformation [1] should be used to properly attack this model. Also, the details of transformation smoothed inference are missing, e.g., what transformations are used? how many transformations are used? (2) In section 4.1, there is a paragraph named “comparison to semi-supervised learning”. Nonetheless, I am pretty confused about the discussion there. First of all, I am pretty confused about what comparisons are conducted there? e.g., is your method used more images as in [15,16]. The only useful information I found is “Compared to the semi-supervised learning methods, RoCL takes about 1/4 times faster with the same computation resources”, but how about comparisons on other metrics, e.g., robustness, accuracy? Also, the authors claim that “ours acquires sufficiently high clean accuracy and robustness after 500 epochs (Fig. 3(c)) which takes 25 hours with two RTX 2080 GPUs”. This information is not very useful as no comparisons are provided, e.g., it is possible that [15,16] also get converged within 500 epochs. The authors should carefully ablate the comparison to semi-supervised learning and polish the corresponding descriptions in the main paper. (3) Why your model is trained with 16/255, as all other supervised methods are trained with 8/255. What is your model performance when trained with 8/255? Also, when you do the black-box attack, what if your source model is RoCL? If RoCL generated adversarial examples cannot transfer well to TRADES and AT, maybe you cannot say your model is better on defending against black-box attacks, as such result can only suggest features learned by supervised methods and unsupervised methods are different? (4) Some important experiment setups are missing. For example, how many attack iterations are performed in your attack during training and testing? (5) Minor: one thing the authors highlight in the paper is that the adversarial perturbations used in RoCL are instance-wise. Nonetheless, I think adversarial perturbations (by default) are instance-wise (e.g., they usually cannot transfer to other images, except some additional tricks are applied to craft universal perturbations) regardless of your learning framework is supervised or unsupervised? Highlight this (default) property is very confused unless some special reasons are provided in the paper? [1] Athalye, Anish, Nicholas Carlini, and David Wagner. "Obfuscated gradients give a false sense of security: Circumventing defenses to adversarial examples." arXiv preprint arXiv:1802.00420 (2018).

Correctness: Yes

Clarity: Some parts are unclear

Relation to Prior Work: Yes

Reproducibility: No

Additional Feedback: Overall, I think it is a reasonably good paper, but the authors should carefully address the above concerns during the rebuttal.


Review 2

Summary and Contributions: Summary: The paper introduces self-supervised contrastive learning to the existing framework of adversarial attack and adversarial learning. Contributions: 1. The paper presents a method to generate adversarial perturbations without label information. 2. The paper unifies adversarial learning and self-supervised learning.

Strengths: 1. It is valuable to study adversarial attacks and adversarial learning in an unsupervised setting. 2. The proposed method is a sensible approach to achieve adversarial robustness without labels.

Weaknesses: 1. Some experimental results do not support the authors’ claim on the effectiveness of the proposed method, or require further explanation. 2. The proposed linear evaluation of RoCL is a very standard adversarial learning technique and the significance of the proposed transformation smoothed inference is limited.

Correctness: The method and experimental settings are correct. Some claims should be further explained.

Clarity: The paper is well written. Some details on the experiment could be provided for better clarity.

Relation to Prior Work: The authors clearly explain the difference between the proposed method and the cited papers. However, it does not mention the following work which also adopts self-supervised learning for adversarial robustness. A methodological and empirical comparison should be made. Naseer, Muzammal, et al. "A Self-supervised Approach for Adversarial Robustness." CVPR. 2020.

Reproducibility: Yes

Additional Feedback: 1. In Eqs. (1) and (4), why the adversarial attack is found by subtracting the gradient from the current x? As we are maximizing the loss, shouldn’t the definition be $x^{i+1} = \Pi_B(x^i+\alpha \sign(Grad)))$ for Eq. (1) and similarly for Eq. (4)? 2. Eq. (7) proposed for linear evaluation of RoCL is simply the objective function of the standard adversarial learning (Ref [9]). Eq. (8) proposed for transformation smoothed classifier seems to take the majority vote of predictions among different augmentations. The authors may consider further explaining the novelty of these two contributions. 3. Some quantitative results in Table 1 seem inconsistent with the claimed benefits. a) Line 230: RoCL achieves higher robustness against $ell_infty$ attacks; In Table 1, accuracy of RoCL is lower than AT and TRADES for seen attacks. b) Line 239: RoCL achieves higher robustness against black-box attachs; In Table 2, accuracy of RoCL is lower than AT for black-box attacks generated via TRADES. c) Line 249: Models fine-tuned with the proposed method obtain better robustness and higher clean accuracy over models trained from scratch; In Table 1, RoCL+AT and RoCL+TRADES do not always outperform AT and TRADES, respectively, both in terms of robustness and clean accuracy. Some details may be further explained. d) A formula for self-supervised loss, used in the method RoCL+AT+SS, should be provided for clarity. e) What is the information behind between colors in Figs. 3(a) and 3(b)? f) Result for attack target image ablation (Line 291) refers to Table 4; the purpose of Table 5 is not mentioned in the paper. Based on the above comments, I vote for a rejection now, but I want to see the author response and may adjust the overall score accordingly. -------------------------------------------------------------------------------------------------------------------------------------------- After authors feedback: As the authors have addressed my concerns, I decided to upgrade my score to 6.


Review 3

Summary and Contributions: The paper proposes a novel framework for learning adversarially robust deep network representations without using any labels. Specifically, the proposed framework involves an unsupervised contrastive based instance discrimination model (e.g. SimCLR) which is coupled with label-free instance-wise adversarial attacks that make the model confuse the instance classification task. So, to unsupervised learn adversarially robust representations, the instance discrimination model is trained in an adversarially robust way by exploiting instance-wise adversarial attacks. The authors evaluate the adversarial robustness of the learned representations by training linear classifiers on them and demonstrate that, although the representation are adversarially learned without any labels, in many cases they are comparable or better (in terms of robustness) than state-of-the-art fully-supervised adversarial methods while at the same time they have better classification accuracy on clean images.

Strengths: + What I found very interesting in this paper is that the proposed method, despite not using labels during representation learning, in many cases (e.g., in unseen or black box attacks) is better than state-of-the-art fully supervised adversarial learning methods (i.e., AT [9], TARDE [2]) while also achieving better classification accuracy on clean images. So, in many cases this unsupervised method is a better alternative than supervised methods for achieving adversarial robustness! Therefore, I believe this unsupervised adversarial learning idea, which, to best of my knowledge, it is proposed for the first time here, is a significant contribution to the field that it will probably attract further interest in the future. + Detailed experimental analysis in various settings! + The authors provide the source code.

Weaknesses: (W1) I think the authors should be more carefully describe their contributions in the abstract and introduction (e.g., in  in lines 5-6, 38-39, and 47). They overemphasize that the proposed method is able to adversarially learn robust neural networks without any labels or in a fully-unsupervised manner. Although I understand what they mean, I believe that it is not expressed rigorously enough since, at the end, the proposed method still needs to use labels in order to train the linear classifiers. Because of that it can be confusing to the reader as well. I found much better the way the contribution is stated in the first sentence of the conclusion (i.e., adversarially learn robust representations / features without labels) and I would advise to use it in the introduction and abstract as well. In general, they should make it more clear that the proposed method does not need any labels for the learning of adversarially robust features but still requires labels for the downstream task (e.g., classification). Also, this distinction should be made more clear when comparing with [15] and [16] in related work (lines 89-91).  (W2) The description of the methodology in section 3.1 is somewhat confusing. The purpose of the adversarially learning objective of equation (5) is not clear, since the objective that is actually minimized is that of equation (6). Furthermore, the description is somewhat incomplete, since in Algorithm 1* that they provide in the supplementary, it is revealed that there is also another (regularization) term in the objective. Although this extra regularization term seems to play an important role (see results in Tables 10 and 11 of supplementary) it is not mentioned in the main paper and the authors do not provide in the supplementary any insight for why using. Also, the method seems to be a bit sensitive to the weight labmda of this extra regulization term. *: BTW, I strongly advice to move Algorithm 1 in the main paper; it would make reading much easier. (W3) The description of the experimental results is not in some cases accurate. For instance: - In line 230: "RoCL achieves high robustness against the target attacks l_{inf} ... outperforming supervised adversarial training by Madry et al. [9] and obtaining comparable performance to TRADES [2]". Actually, RoCL (without rLE) is worse than AT[9] for seen attacks (l_{inf}). Also, when compared to TRADES [2] (for seen attacks), the performance gap is quite big to be considered "comparable". Except if the authors mean that RoCL+rLE is better than AT and comparable to TRADES, which is (kind of) true for ResNet18 but not for ResNet50, but then you should fix the typo (i.e., missing +rLE) and be more specific. - In lines 237-238: "Moreover, RoCL obtains much better clean accuracy, and significantly higher robustness over the supervised adversarial learning approaches against unseen types of attacks and black box attacks" Actually for l_2 with e=0.5 attacks, RoCL is worse than AT and TARDE. Also, for black box attacks, RoCL is worse than AT for TRADES attacks. - Similarly, describe more carefully the results in the self-supervised+fine-tuned section of Table 1.

Correctness: They seem correct to me.

Clarity: Except from the section 3.1 (see weakness point (W2) above), the paper is well written.

Relation to Prior Work: The related work section is rich and well written. Good job!

Reproducibility: Yes

Additional Feedback: Overall, I believe the strengths of the paper (i.e., interesting work, solid contribution to the field) outweigh its weakness (which many of them can be easily fixed). Al Clarification: In Table 3 it is not clear if RoCL and Transfer [38] use the same network backbones. If they do not, then the comparison is not fair. Minor fixes: - Describe what A_nat is on the tables. - L249: "which show that the models fine-tuned with our method" ==> "which show that the models *pretrained* with our method" - "takes about 1/4 times faster" ==> "is about 4 times faster" - Please describe what is the self-supervised loss (SS) that you mention in the experiments. Add a citation. - Table 1 is too dense (i.e., it includes too many results) making it hard to follow it and digest it. Consider breaking it into smaller tables or somehow simplifying it. AFTER THE REBUTTAL: The authors answered my comments on their feedback. So, after considering the other reviews, the rebuttal, and the discussion, I decided to keep my initial positive rate (accept).


Review 4

Summary and Contributions: In this paper, the authors propose a method to learn robust representations from unlabeled data. The method, dubbed RoCL, extends instant-wise contrastive learning framework (in this case SimCLR model) with a min-max formulation for adversarial learning. The robust features are learned by maximizing the similarity between an image sample and its instancewise adversary. The authors show results on CIFAR-10/100 datasets on different settings. RoCL achieve comparable robustness to supervised adversarial learning approaches (without using any labels) and improve on unseen type of attacks.

Strengths: + The idea of learning robust features in an unsupervised setting is novel and unexplored. + The proposed approach is simple and easy to understand

Weaknesses: Although I find Sections 1-3 nicely written and easy to follow, I have many issues with the experimental section. There are way too much information (and way too less description). I found Table 1 almost impossible to parse, even going through it multiple times. Much of the experimental setup is not really described either. - How was the the proposed method compared to supervised adversarial training [9,2] on the linear evaluation setting? Do the supervised methods are trained, then the CNN features frozen, then a new linear layer trained on the top of it? - I find the actual task of linear evaluation for adversarial training not very well motivated. The fact of evaluating self-supervised/unsupervised features with a linear probing makes (a bit) of sense, since we want to see how good the learning of features are. However, in the case of feature robustness (which is about the performance of features in a particular task) does not make sense, I dont understand why one would be interested in linear evaluation. I feel like the finetuning scenario makes much more sense -- and unfortunately, only a small part of experiments deal with it. - There are way too much training details missing in the paper. For example, what kind of data augmentation were used on the contrastive learning part? How comes SimCLR works only assuming a batch size of 256 (and therefore a very small number of negative samples at each iteration)? SimCLR require a very large batch size (order of thousands) to make it work. - It is not clear to me why is it necessary to consider both t'(x) and t'(x)^{adv} as positive samples in RoCL. Why the two positive samples instead of just the the adversarial pertubation? Some ablation studies explaining some model choices would be also helpful.

Correctness: - As mentioned above, the algorithmic description of the model seems correct, but there are too much missing information on the experimental section to actually judge its correctness. - The transfer learning studied (CIFAR-10 to CIFAR-100 or vice-versa) might not be the best example of TL, given the task is the same (classification, albeit different number of categories) and the data distribution is still fairly similar.

Clarity: Section1-3 are well written and clear, but I find Section 4 (experiments) not clear at all. There are a lot of missing experimental details (a little bit is described on supp material, but still much is missing).

Relation to Prior Work: Yes.

Reproducibility: No

Additional Feedback: ============== Post-rebuttal comments ============== After seeing the other reviewers' comments and the authors' rebuttal, I decided to upgrade my score to 6. The author answered most of my concerns, but I still find that experimental section could be improved (in particular the clarity and the TL experiments). I would strongly recommend the authors to update the manuscript with the reviewers concerns.

[Author Response · NeurIPS 2020]

We thank all the reviewers for their constructive comments. We are encouraged that reviewers found our method novel
(R1, R3, R4), simple yet effective (R3, R4), the experimental results to be encouraging and thorough (R1, R3), and the
paper well written (R2, R3). We address the individual comments from each reviewer below.

**[Reviewer #1] Comments regarding transformation smoothed inference.** We first
want to emphasize that RoCLs work well without smoothed classifiers (RoCL, RoCL
+rLE). Nonetheless, we tested transformation smoothed classifier (sampled 30 times, with
random-fixed size crop, random color distortion) against EoT as suggested, and observed
degradation of performance on CIFAR-10 with ResNet-18 **(33.24%)**. This is still **robust**,
compared with other randomized defenses (0.03%) [21]. Also, please note that RoCL
+ smoothed can still defend against black box attack (+9.8%, Fig.3(d)) and gain clean
accuracy (+0.5%) under the trade-off with white-box EoT accuracy. Moreover, RoCL
w/o smoothed classifier is robust against EoT attack **(37.28%)**. We thank you for your
insightful suggestion and will clarify this. **What comparison is in "Comparison to**
**semi-supervised learning"?** We intended to discuss the time-efficiency of RoCL in that
section, and will revise the title to "Training efficiency∼" for improved clarity. We further report the training time for
ResNet18 trained on CIFAR10 to reach convergence for the two methods, with two RTX 2080 Ti GPUs (Table B).
**What if you use RoCL as the source model for the blackbox attacks?** Adversarial examples generated from RoCL
with instance-wise attack and PGD attack both succeed in attacking AT and TRADES. (Table B; epsilon=8/255).
**Why train models with 16/255? The number of iterations?** This seems like a misunderstanding since as described
in L436, we trained all models including ours with 8/255 with 7 iterations. We used 20 iterations for test (L424,L442).
**Conventional adversarial perturbations are instance-wise.** We will rename our attack as "instance-identity attack".

**[Reviewer #2] Novelty of rLE and transformation smoothed classifier.** Our main contributions are 1) the proposal
of a **novel adversarial perturbation** which makes the model to confuse a sample to another, 2) the **contrastive**
**learning** framework for **unsupervised adversarial learning**. We proposed the rLE as an evaluation measure for
robustness of unsupervised adversarial learning, and t-smoothed inference an alternative for existing smoothed classifier,
but they are rather technical details and we do not claim them as our contributions (Please see L60-67 for our claims).
**Inconsistency between claimed benefits in certain lines and Table 1.** We will fix the inaccurate descriptions as
follows: (a) We will rename RoCL to RoCL+rLE, which does outperform AT and obtain comparable performance to
TRADES. (b) We will clarify that RoCL achieves higher robustness against "AT black-box attacks", as described in
L262-265. (c) We will revise this as RoCL + finetuning "sometimes" outperform models trained from scratch.
**Colors in Fig.3(a,b)** The markers in Fig.3 with different shapes denote instances belonging to different instances, and
the green and red color denote clean and adversarial (L286) instances, respectively. We will clarity this in the revision.
**Purpose of Table 5.** Table 5 shows that RoCL remains robust even with an increased number of attack iterations.
**Missing formula for RoCL+AT+SS, typos in Eq.1 and Eq.4, a missing reference** Thank you for pointing them out.
We used both AT loss and Eq.3 for RoCL+AT+SS. We will fix the typos ("-") in Eq.1 and Eq.4 to "+", and cite and
discuss [Naseer et al. 20] which is different from ours that proposes purifier network trained with Euclidean adversary.
**[Reviewer #3] The methodology in section 3.1 is confusing.** Here we generate adversarial examples using Eq.5
which fools the instance identity, and explicitly train the model using Eq.6 (L169-172) using the generated adversarial
examples. The regularization term in the Algorithm 1 yields small gain (+1.07%) on robustness, and the most important
loss is $\mathcal{L}_{con}$ (L187-193). We will include the descriptions of the regularizer and the Algorithm in the main paper.
**"Unsupervised" could be misleading since label is used for downstream tasks.** We will revise the texts to clarify
that we do not require any lables to learn adversarially robust representations, but need labels for downstream tasks.
**The description of the experimental results should be specific.** We apologize and will revise the inaccurate descrip-
tions. Please see the comments to Reviewer #2 (Inconsistency between claimed benefits and Table 1).
**What is the backbone of [38]?** [38] used WRN 32-10 (L446), which has much larger number of parameters and
depths compared to ResNet18 used by our model that outperforms [38]. This demonstrates the effectiveness of RoCL.
**[Reviewer #4] Do you need new linear layers on top of supervised [9,2] to compare against RoCL?** Since super-
vised methods already have a linear layer at training time, it is unnecessary to add an additional linear layer.
**Why use linear evaluation? Finetuning makes more sense.** Since linear evaluation "freezes" the representation, it
is the most direct way to measure the robustness of the learned representations (L196-198). Finetuning (L247-253)
will change the lower layer representations and will make it difficult to separate the effect of supervised adversarial
finetuning from effect of RoCL. **Which data augmentation were used?** As described in L147-148,173, we used
random crop, and random color distortion. **Batchsize 256 too small.** SimCLR [12] experimented on ImageNet, and
we empirically found that bathsize of 256 is sufficient for CIFAR-10 and CIFAR-100. Note that we use adversarial
examples as additional positive examples as well. **Why consider both $t'(x)$ and $t'(x)^{adv}$ as positive samples?** Using
both of them is essential since we are targeting for both clean and adversarial accuracies.
**Transfer learning setup might not be the best example of TL.** We agree. However, since the baseline [38] is using
the described experimental setup, we had to follow it for a fair and direct comparison (L449).

Table A: Training cost

|         | Ours  | Semi[15] |
|---------|-------|----------|
| Dataset | **50K** | 150K   |
| Time    | **41.7h** | 66.7h |
| Epoch   | 1000  | 200      |

Table B: Blackbox attack

|        | RoCL Source | |
|--------|------------|------|
| Target | our attack | PGD  |
| AT     | **42.87**  | 69.13 |
| Trades | **41.59**  | 64.81 |

[Meta-Review · NeurIPS 2020]

This paper is a first work that successfully shows we can learn robust models in an unsupervised manner which is a significant contribution to the field. The proposed approach is simple and easy to understand and the empirical results are pretty encouraging. Reviewers had concerns regarding the writing and the experimental setup, but most concerns were addressed in the rebuttal. I would rank the paper as a solid poster and encourage the authors to improve the manuscript following the reviewer's advice for the camera ready.